# Quantifying epidemiological drivers of *gambiense* human African Trypanosomiasis across the Democratic Republic of Congo

**Ronald E. Crump**[1,2,3☯]\*, **Ching-I Huang**[1,2☯], **Edward S. Knock**[1,4], **Simon E. F. Spencer**[1,4], **Paul E. Brown**[1,2], **Erick Mwamba Miaka**[5], **Chansy Shampa**[5], **Matt J. Keeling**[1,2,3], **Kat S. Rock**[1,2]

**1** Zeeman Institute for System Biology and Infectious Disease Epidemiology Research, The University of Warwick, Coventry, United Kingdom, **2** Mathematics Institute, The University of Warwick, Coventry, United Kingdom, **3** The School of Life Sciences, The University of Warwick, Coventry, United Kingdom, **4** The Department of Statistics, The University of Warwick, Coventry, United Kingdom, **5** Programme National de Lutte contre la Trypanosomiase Humaine Africaine (PNLTHA), Kinshasa, D.R.C.

☯ These authors contributed equally to this work.
\* r.e.crump@warwick.ac.uk

**Data Availability Statement:** Data cannot be shared publicly because they were aggregated from the World Health Organisation's HAT Atlas

## Abstract

*Gambiense* human African trypanosomiasis (gHAT) is a virulent disease declining in burden but still endemic in West and Central Africa. Although it is targeted for elimination of transmission by 2030, there remain numerous questions about the drivers of infection and how these vary geographically.

In this study we focus on the Democratic Republic of Congo (DRC), which accounted for 84% of the global case burden in 2016, to explore changes in transmission across the country and elucidate factors which may have contributed to the persistence of disease or success of interventions in different regions. We present a Bayesian fitting methodology, applied to 168 endemic health zones (∼100,000 population size), which allows for calibration of a mechanistic gHAT model to case data (from the World Health Organization HAT Atlas) in an adaptive and automated framework.

It was found that the model needed to capture improvements in passive detection to match observed trends in the data within former Bandundu and Bas Congo provinces indicating these regions have substantially reduced time to detection. Health zones in these provinces generally had longer burn-in periods during fitting due to additional model parameters.

Posterior probability distributions were found for a range of fitted parameters in each health zone; these included the basic reproduction number estimates for pre-1998 ($R_0$) which was inferred to be between 1 and 1.14, in line with previous gHAT estimates, with higher median values typically in health zones with more case reporting in the 2000s.

Previously, it was not clear whether a fall in active case finding in the period contributed to the declining case numbers. The modelling here accounts for variable screening and suggests that underlying transmission has also reduced greatly—on average 96% in former Equateur, 93% in former Bas Congo and 89% in former Bandundu—Equateur and

which is under the stewardship of the WHO. Data are available from the WHO (contact neglected. diseases@who.int or visit https://www.who.int/ trypanosomiasis_african/country/foci_AFRO/en/) for researchers who meet the criteria for access to confidential data.

**Funding:** This work was supported by the Bill and Melinda Gates Foundation (www.gatesfoundation. org) through the Human African Trypanosomiasis Modelling and Economic Predictions for Policy (HAT MEPP) project [OPP1177824] (C.H, R.E.C, P. B, K.S.R. and M.J.K.) and through the NTD Modelling Consortium [OPP1156227, OPP1186851 and OPP1184344] (K.S.R., E.K., S.E. F.S and M.J.K.). The funders had no role in study design, data collection and analysis, decision to publish, or preparation of the manuscript.

**Competing interests:** The authors have declared that no competing interests exist.

Bandundu having had the highest case burdens in 2000. This analysis also sets out a framework to enable future predictions for the country.

## Author summary

*Gambiense* human African trypanosomiasis (gHAT; sleeping sickness) is a deadly disease targeted for elimination of transmission by 2030, however there are still several unknowns about what factors influence continued transmission and how this changes with geographic location.

In this study we focus on the Democratic Republic of Congo (DRC), which reported 84% of the global cases in 2016 to try and explain why some regions of the country have had more success than others in bringing down case burden. To achieve this we used a state-of-the-art statistical framework to match a mathematical gHAT model to reported case data for 168 regions with some case reporting during 2000–2016.

The analysis indicates that two former provinces, Bandundu and Bas Congo had substantial improvements to case detection in fixed health facilities in the time period. Overall, all provinces were estimated to have reductions in (unobservable) transmission including ∼96% in former Equateur. This is reassuring as case finding effort has decreased in that region.

The model fitting presented here will allow predictions of gHAT under alternative intervention strategies to be performed in future studies.

## Introduction

*Gambiense* human African trypanosomiasis (gHAT) is a disease caused by the protozoan parasite *Trypanosoma brucei gambiense* which is transmitted by tsetse. The disease has two distinct stages during which the disease progresses from mild to severe, and can lead to death without treatment.

gHAT occurs throughout Western and Central Africa, with 15 countries reporting new cases in the period 2000–2016 [1]. The majority of the detected gHAT cases are in the Democratic Republic of Congo (DRC) where 84% of the new cases in 2016 were reported [2]. While these cases are predominantly in the former province of Bandundu, they are widespread across this large country (230 of 516 health zones had reported cases between 2012 and 2016).

It has long been understood that treatment of gHAT patients not only prevents excess mortality but it can also reduce the time spent infectious, and thereby reduce onward transmission in the population. A combination of active screening and passive surveillance followed by treatment of cases has resulted in a decline in the number of new cases from 25,841 (16,951 in DRC) in 2000 to 2,110 (1,768 in DRC) in 2016 [1]. During this time period both diagnostics and drugs for gHAT have evolved with vast improvements for patients.

Traditional active screening of at-risk populations is done by mobile teams visiting villages and performing an initial mass screen using serological tools (usually the card agglutination test for trypanosomes—CATT), followed by microscopy for serologically positive suspects to confirm presence of the parasite. This microscopic parasitological confirmation of a case was required before drug administration. For the treatments available in the previous two decades, a final "staging" test—a lumbar puncture to establish whether a patient has trypanosomes or elevated white blood cell count in cerebrospinal fluid—was required to select appropriate

treatment. This necessary, multi-step diagnostic pathway currently precludes the possibility of mass drug administration as used in control programmes for other neglected tropical diseases (NTDs).

For infected people who evade detection by active screening due to imperfect diagnostics, non-attendance in active screening, or whose village is not screened, it is possible for them to self-present at fixed health facilities and be diagnosed through passive surveillance. Not all fixed health facilities have gHAT diagnostics but this has been improved over the last 20 years. Globally, WHO estimate that in 2012, 41%, 71%, and 83% of at-risk population lived within 1, 3 and 5 hours of health facilities with these diagnostics respectively [3], and by 2017 this increased to 58%, 79%, 89% [1]. It is not clear what quantitative impact this improvement has had on time to detection.

Along with many other NTDs, gHAT is the subject of two World Health Organisation goals; for gHAT these are (1) elimination as a public health problem by 2020 and (2) elimination of transmission (EOT) by 2030. If we are to truly strive to reach this second elimination of transmission goal, then it is of utmost importance to understand and quantify the reasons for success to date (indeed, it is expected that we are on track for the first goal [1] with some countries identified as eligible for validation of gHAT elimination as a public health problem in the latest WHO report [4]), and identify what factors may have hindered progress.

gHAT transmission is known to be highly focal—now burden of disease is decreasing globally, there remain pockets of infection in geographically disconnected areas [1]. Previous mathematical modelling work has shown that, whilst persistence of infection at very low prevalences is generally surprising for infectious diseases, the slow progression of gHAT in individuals enables this infection to remain extant for long periods of time in small settlements [5]. Other transmission modelling has examined drivers of gHAT in specific foci and concluded that, by fitting to longitudinal human case data, there must exist heterogeneity in risk of humans populations both in terms of exposure to tsetse and also in participation in active screening [6, 7]. Other factors which are likely to vary geographically include access to fixed health facilities with gHAT diagnostics—this can impact the time individuals spend infected and the risk that they die without diagnosis—and the regional density of tsetse. At present it is, however, unknown which drivers are influencing transmission in different regions.

In the present study we consider what epidemiological variables are driving transmission across the health zones ($\sim$ 100,000 population size) of DRC using a dynamic transmission model, and examine how control interventions from 2000 to 2016 have impacted infection and modified some of these variables over time. To achieve parameter estimation across the country we utilise an automated Bayesian fitting procedure with an adaptive Metropolis-Hastings random walk and in-built convergence diagnostics. The samples from the posterior probability distributions of fitted parameters from this method can be used to examine within health zone parameter averages and uncertainty as well as comparing estimates across health zones. The mechanistic model can be used to infer the level of transmission to humans over time even though it is not directly observable in data; the posterior parameters are used to do this.

There are three major outputs from this study. Firstly, the parameterisation of our gHAT transmission model which allows future predictions to be made considering different gHAT intervention strategies [8]. With future predictions based on realistic, localised epidemiological parameters; economic evaluations of the cost-effectiveness of different approaches to the control of gHAT are possible [9]. Secondly, the evaluation of the effectiveness of interventions against gHAT in DRC across the period 2000 to 2016. Lastly, but potentially of most value to our research going forwards, the provision of a rapid, repeatable model fitting framework to facilitate future research around the model presented here and other variants of it.

## Materials and methods

### Data

Data on gHAT cases in DRC were obtained from the WHO who curate the global HAT Atlas database [1, 10, 11]. The data in the HAT Atlas are case data aggregated by location, year and surveillance type. Location was defined by the available geolocation and geographical identifier information, while surveillance type was either active or passive screening. There were 117,573 records in the HAT Atlas data file covering the period 2000–2016. Although many data entries for 2015 and 2016 have information of the stage (1 or 2) of disease, very little staging information was available prior to 2015.

Because administrative areas may be redefined or renamed over time, the HAT Atlas records were matched to a single, recent map obtained as a shapefile (UCLA, personal communication). Matching to the shapefile was performed by geolocation, where available, plus an identifier matching both to geolocated locations and directly to the administrative areas stored in the shapefile. Other geographical data were sourced from the Humanitarian Data Exchange —an older United Nations Office for the Coordination of Humanitarian Affairs (OCHA) health zone shapefile, and geolocations of localities (OCHA) and health facilities (Global Healthsite Mapping Project) within DRC. The additional geographical data provided alternative names for administrative/health areas as well as names and geolocations of locations and hence facilitate the data cleaning and matching process. While geolocations (longitude and latitude) were available for most records they may not be reliable indicators of position as they may have been assigned as the centroid of the lowest available level administrative/health area if not recorded directly. A standardisation procedure was performed on area and location names from all sources. Sequential matching was then carried out, with unmatched records being carried forward to the next step. The location and health site lists were combined and located on the UCLA shapefile to obtain administrative area names consistent with this map. Detail on this process is available in the Supporting Information (S1 Text).

Following data matching, the data were aggregated within health zone, year and surveillance type to produce health zone level data sets for all health zones in which cases were reported. It is noted that for a small number of health zone–year combinations the reported number of cases detected by active screening was more than the declared number of people screened, which was zero in some instances.

Cases of gHAT are not recorded across all of the DRC; for many regions this is because they have not historically observed gHAT cases and are not believed to be endemic for gHAT, but for others it is due to challenging accessibility. Fig 1 shows the regions where there were no data as well as the status of the analysis performed in the present study. Inference was not performed for all health zones where there were data available; either because there were too few observations (Data < 10), no cases were detected (No Detections) or because transmission is believed not to be taking place, since there is an absence of tsetse habitat (No Transmission). Inference was deemed to be possible where there were 10 or more observations, where an observation was an aggregate record for a single year relating to either more than 20 people being actively screened or cases detected passively.

Fig 2 contains three maps of aggregated numbers of new cases by health zone for three non-consecutive five-year periods spanning the period 2000–2016 for which data were available. These maps illustrate the decrease in cases reported over time, in particular in the former provinces of Equateur (in the North-West of DRC), which had the highest burden in 2000, and Bandundu (situated to the east of Kinshasa), which despite its decrease is now the most highly-endemic region globally. In the North there was an increase in reported cases due to the presence of Médecins Sans Frontières (MSF) who use an alternative diagnostic algorithm

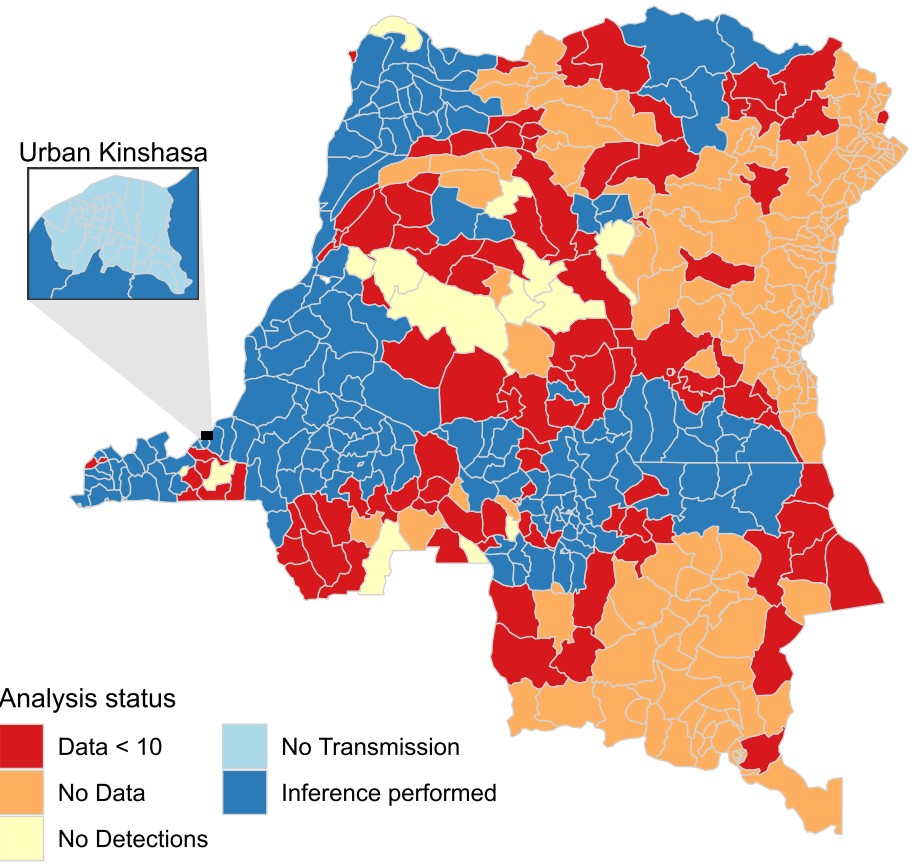

**Fig 1. Status of the individual health zone-level analyses, if the analysis was not run the reason is indicated.**
Shapefiles used to produce these maps are available under an ODC-ODbL licence at https://data.humdata.org/dataset/drc-health-data.

to that used by the national programme. The MSF algorithm did not feature parasitological confirmation after serological testing, and hence had a higher sensitivity but lower specificity (similar to those described in [12]). Some cases were reported in urban Kinshasa (Fig 2), however these are believed to be the result of infections that occurred elsewhere and there is

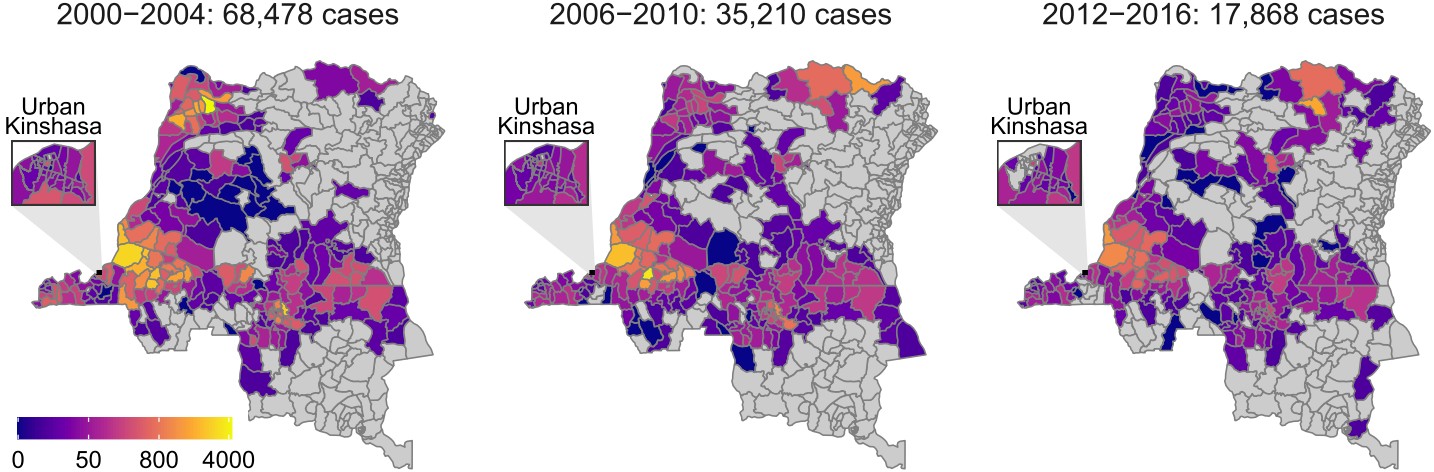

**Fig 2. Total number of new cases recorded within health zone and five-year period.** Shapefiles used to produce these maps are available under an ODC-ODbL licence at https://data.humdata.org/dataset/drc-health-data.

assumed to be no transmission within urban Kinshasa (Fig 1) and hence no model fitting was performed for those health zones.

## Model

**gHAT infection model.** The deterministic gHAT model equations are ordinary differential equations given in Eq (1) and correspond to the model schematic presented in Fig 3. The model is the variant "Model 4" of that presented elsewhere [6, 7, 13–16]: in this model variant human hosts are assumed to be either at low-risk and randomly participate in screening (subscript $H1$), or high-risk and never participate in screening (subscript $H4$). Eq (1) therefore uses indices to denote the risk classes, using only $i \in \{1, 4\}$, with other subscripts being reserved for model variants with high-/low-risk structures as presented in other publications [6, 7] but not considered here. In Model 4, tsetse bites are assumed to be taken on humans or non-reservoir animals, however, the non-reservoir animal species do not need to be explicitly modelled. Infection occurring in animals is not modelled in the present study.

For simplicity we consider a closed population of size $N_H$ individuals, with natural mortality and births. Furthermore, we assume that any deaths related to gHAT will be replaced by new susceptibles entering the population, therefore the parameter $\gamma_H$ represents a mix of disease-induced deaths and detection in stage 2. To compute the cases found in stage 2 passive detection

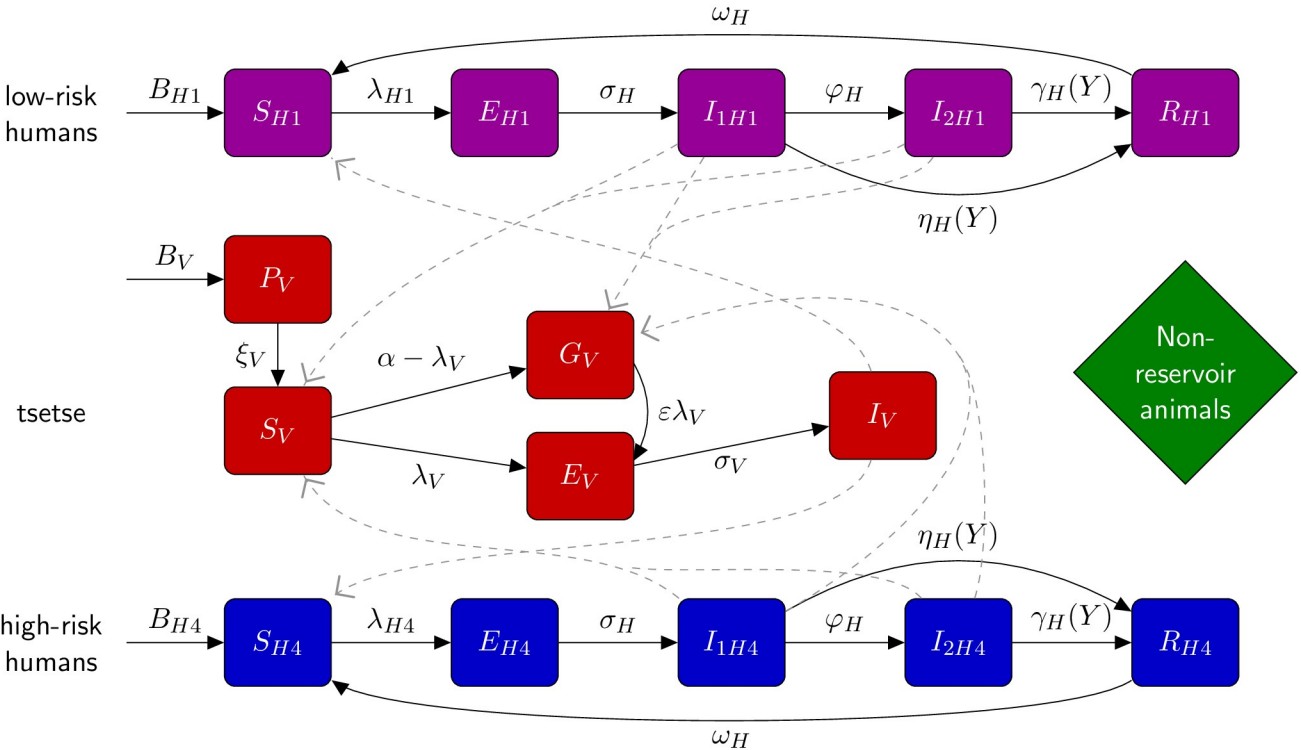

**Fig 3. Illustration of compartmental gHAT model.** Multi-host model of gHAT with one host species able to confer *T. b. gambiense* (humans), a further non-reservoir species and tsetse. After a short incubation period, infected human hosts follow the progression which includes an infectious stage 1 disease, $I_{1H}$, infectious stage 2 disease, $I_{2H}$, and a hospitalised/recovering class, $R$. Pupal stage tsetse, $P_V$, emerge into unfed adults. Unfed tsetse are susceptible, $S_V$, and following a blood-meal become either exposed, $E_V$, or have reduced susceptibility to the trypanosomes with subsequent bloodmeals, $G_V$. Tsetse select their blood-meal from one of the host types dependant upon innate feeding preference and relative host abundance. High-risk humans are *r*-fold more likely to receive bites than low-risk humans. Any blood-meals taken upon non-reservoir animals do not result in infection. The transmission of infection between humans and tsetse is shown by grey paths. This figure is adapted from the original model schematic [6].

we multiple the reporting rate $u$ by the stage 2 exit rate $\gamma_H$. As we only expect disease-induced deaths from stage 2 infection, the same does not apply to the stage 1 detection rate.

The model is parameterised with a combination of fixed and fitted parameters. Fixed parameters (see Table 1) generally correspond to assumed biological values that are unlikely to vary across the DRC, such as the human mortality rate ($\mu_H$), tsetse bite rate ($\alpha$) and stage 1 to stage 2 disease progression in humans ($\varphi_H$). Fitted parameters (see Table 2) are those which are likely to be correlated with region including the proportion of the population at low-risk of infection ($k_1$), the relative rate at which high-risk humans are bitten by tsetse ($r$), the reporting rate ($u$) (corresponding to access to health facilities with gHAT testing capacity), and parameters linked to the time to passive detection or disease-induced mortality ($\eta_H$ for stage 1 and $\gamma_H$ for stage 2). Some parameters of the model are not fitted themselves but are functions of other parameters. The proportion of high-risk people in the population was calculated as $k_4 = 1 - k_1$. The exit rate from stage 2 pre-1998 was assumed to be less than that achieved post-1998, and was therefore calculated as $\gamma_H^{\text{pre}} = b_{\gamma_H^{\text{pre}}} \gamma_H^{\text{post}}$, where $\gamma_H^{\text{post}}$ was the treatment rate from stage 2 post-1998 and $b_{\gamma_H^{\text{pre}}}$ is a fitted value in the range zero to one. The tsetse-to-human relative density, $m_{\text{eff}}$, is calculated from $R_0$ using the next generation matrix approach.

$$
\text{Humans}
\begin{cases}
\dfrac{dS_{Hi}}{dt} &= \mu_H N_{Hi} + \omega_H R_{Hi} - \alpha m_{\text{eff}} f_i \dfrac{S_{Hi}}{N_{Hi}} I_V - \mu_H S_{Hi} \\[2mm]
\dfrac{dE_{Hi}}{dt} &= \alpha m_{\text{eff}} f_i \dfrac{S_{Hi}}{N_{Hi}} I_V - (\sigma_H + \mu_H) E_{Hi} \\[2mm]
\dfrac{dI_{1Hi}}{dt} &= \sigma_H E_{Hi} - (\varphi_H + \eta_H(Y) + \mu_H) I_{1Hi} \\[2mm]
\dfrac{dI_{2Hi}}{dt} &= \varphi_H I_{1Hi} - (\gamma_H(Y) + \mu_H) I_{2Hi} \\[2mm]
\dfrac{dR_{Hi}}{dt} &= \eta_H(Y) I_{1Hi} + \gamma_H(Y) I_{2Hi} - (\omega_H + \mu_H) R_{Hi}
\end{cases}
$$

$$
\text{Tsetse}
\begin{cases}
\dfrac{dP_V}{dt} &= B_V N_H - (\xi_V + \frac{P_V}{K}) P_V \\[2mm]
\dfrac{dS_V}{dt} &= \xi_V \mathbb{P}(\text{pupating}) P_V - \alpha S_V - \mu_V S_V \\[2mm]
\dfrac{dE_{1V}}{dt} &= \alpha(1 - f_T(t)) p_V \left(\sum f_i \dfrac{(I_{1Hi} + I_{2Hi})}{N_{Hi}} + f_A \dfrac{I_A}{N_A}\right)(S_V + \varepsilon G_V) \\
&\quad - (3\sigma_V + \mu_V + \alpha f_T(t)) E_{1V} \\[2mm]
\dfrac{dE_{2V}}{dt} &= 3\sigma_V E_{1V} - (3\sigma_V + \mu_V + \alpha f_T(t)) E_{2V} \\[2mm]
\dfrac{dE_{3V}}{dt} &= 3\sigma_V E_{2V} - (3\sigma_V + \mu_V + \alpha f_T(t)) E_{3V} \\[2mm]
\dfrac{dI_V}{dt} &= 3\sigma_V E_{3V} - (\mu_V + \alpha f_T(t)) I_V \\[2mm]
\dfrac{dG_V}{dt} &= \alpha(1 - f_T(t)) S_V \\
&\quad - \alpha\left(f_T(t) + (1 - f_T(t)) p_V \varepsilon \left(\sum f_i \dfrac{(I_{1Hi} + I_{2Hi})}{N_{Hi}} + f_A \dfrac{I_A}{N_A}\right) G_V\right) \\
&\quad - \mu_V G_V
\end{cases}
\tag{1}
$$

**Table 1. Model parameterisation (fixed parameters).** Notation, a brief description, and the values used for fixed parameters.

| Notation | Description | Value | |
|---|---|---|---|
| $N_H$ | Total human population size in 2015 | Fixed for each health zone | [17] |
| $\mu_H$ | Natural human mortality rate | $5.4795 \times 10^{-5}$ days$^{-1}$ | [18] |
| $B_H$ | Total human birth rate | $= \mu_H N_H$ | |
| $\sigma_H$ | Human incubation rate | 0.0833 days$^{-1}$ | [19] |
| $\varphi_H$ | Stage 1 to 2 progression rate | 0.0019 days$^{-1}$ | [20, 21] |
| $\omega_H$ | Recovery rate or waning-immunity rate | 0.006 days$^{-1}$ | [22] |
| Sens | Active screening diagnostic sensitivity | 0.91 | [12] |
| $B_V$ | Tsetse birth rate (per capita rate of depositing new pupae) | 0.0505 days$^{-1}$ | [13] |
| $\xi_V$ | Rate of pupal development to adult flies | 0.037 days$^{-1}$ | |
| $K$ | Pupal carrying capacity | $= 111.09 N_H$ | [13] |
| $\mathbb{P}(\text{pupating})$ | Probability of a pupa surviving to emerge as an adult fly | 0.75 | |
| $\mu_V$ | Tsetse mortality rate | 0.03 days$^{-1}$ | [19] |
| $\sigma_V$ | Tsetse incubation rate | 0.034 days$^{-1}$ | [23, 24] |
| $\alpha$ | Tsetse bite rate | 0.333 days$^{-1}$ | [25] |
| $p_V$ | Probability of tsetse infection per single infective bite | 0.065 | [19] |
| $\varepsilon$ | Reduced susceptibility factor for non-teneral (previously fed) flies | 0.05 | [6] |
| $f_H$ | Proportion of blood-meals on humans | 0.09 | [26] |
| $\text{disp}_{\text{act}}$ | Overdispersion parameter for active detection | $4 \times 10^{-4}$ | – |
| $\text{disp}_{\text{pass}}$ | Overdispersion parameter for passive detection | $2.8 \times 10^{-5}$ | – |

The value of $B_V$ was chosen to maintain constant population size in the absence of vector control interventions. The value of $K$ was chosen to reflect the observed bounce back rate.

**Table 2. Model parameterisation (fitted parameters).** Notation, brief description, and information on the prior distributions for fitted parameters.

| Notation | Description | Prior distribution[1] | Percentiles of prior distribution [2.5, 50 & 97.5%] | Unit |
|---|---|---|---|---|
| $R_0$ | Basic reproduction number (NGM approach) | $1 + \text{Exp}(10)$ | [1.003, 1.069, 1.369] | - |
| $r$ | Relative bites taken on high-risk humans | $1 + \Gamma(3.68, 1.09)$ | [2.015, 4.654, 10.028] | - |
| $k_1$ | Proportion of low-risk people | $B(16.97, 3.23)$ | [0.6564, 0.8514, 0.9609] | - |
| $\eta_H^{\text{post}}$ [2] | Treatment rate from stage 1, 1998 onwards | $\Gamma(3.54, 5.32 \times 10^{-5})$ | $[4.59, 17.1, 42.9] \times 10^{-5}$ | days$^{-1}$ |
| $\gamma_H^{\text{post}}$ [2] | Combined treatment and disease-induced death rate from stage 2, 1998 onwards | $\Gamma(2.45, 0.00192)$ | $[7.59, 40.7, 121] \times 10^{-4}$ | days$^{-1}$ |
| $b_{\gamma_H^{\text{pre}}}$ | Relative treatment/death rate from stage 2 factor, pre-1998 | $B(1, 1)$ | [0.025, 0.500, 0.975] | - |
| Spec | Active screening diagnostic specificity | $0.998 + (1 - 0.998)B(7.23, 2.41)$ | [0.9989, 0.9995, 0.9999] | - |
| $u$ | Proportion of stage 2 passive cases reported | $B(20, 40)$ | [0.2208, 0.3315, 0.4564] | - |
| $d_{\text{change}}$ [3] | Midpoint year for passive improvement | $2000 + (2017 - 2000)B(5, 6)$ | [2003.2, 2007.7, 2012.5] | Year |
| $\eta_{H_{\text{amp}}}$ [4] | Relative improvement in passive stage 1 detection rate | $\Gamma(2.013, 1.049)$ | [0.258, 1.775, 5.870] | - |
| $\gamma_{H_{\text{amp}}}$ [4] | Relative improvement in passive stage 2 detection rate | $\Gamma(1.001, 5)$ | [0.127, 3.471, 18.455] | - |
| $d_{\text{steep}}$ [4] | Speed of improvement in passive detection rate | $\Gamma(39.57, 0.0270)$ | [0.761, 1.058, 1.424] | years$^{-1}$ |

[1] Where Exp(.), Γ(.) and B(.) are the exponential, gamma (parameterised with shape and scale) and beta distributions, respectively.

[2] Former province-specific priors used for $\eta_H^{\text{post}}$ and $\gamma_H^{\text{post}}$; prior distributions and percentiles for Bandundu presented, see SI "S1 Text" for other former provinces.

[3] $d_{\text{change}}$ is only fitted in the former province of Bandundu.

[4] $\eta_{H_{\text{amp}}}$, $\gamma_{H_{\text{amp}}}$ and $d_{\text{steep}}$ are only fitted in the former provinces of Bandundu and Bas Congo; the prior distributions and percentiles presented relate to Bandundu, see SI "S1 Text" for Bas Congo.

The actual number of vectors is $S_V$, $E_{1V}$, $E_{2V}$, $E_{3V}$, $I_V$ and $G_V$ multiplied by $N_V/N_H$, where $N_V$ is the total population of adult tsetse and $N_H = N_{H1} + N_{H4}$ denotes the total human population. Then, the effective probability of human infection per single infective tsetse bite $m_{\text{eff}}$ is defined as $N_V p_H/N_H$ with the original vector-to-human transmission probability $p_H$.

Whilst the tsetse population size is assumed constant in almost all simulations presented, we use an explicitly host-vector model to enable us to simulate the impact of tsetse interventions. Vector control is included in Eq (1) as $f_T(t)$, the probability of a fly both hitting a tiny target and subsequently dying at time $t$. The value of $f_T(t)$ is dependent on the population reduction achieved by any vector control performed. For Yasa Bonga health zone, the only health zone in which vector control took place prior to the end of the data collection period, a 90% reduction in tsetse population in the first year after biannual deployment of tiny targets was introduced [27]; more details are given in the SI (S1 Text) about the functional form of $f_T(t)$, which was originally presented elsewhere [13]. Other tsetse parameters include the pupal stage $P_V$ from which new, unfed (teneral) adult flies emerge and it is on this pupal class where we place our density-dependent carrying capacity $K$, which governs the bounceback speed of the population in the case where $f_T(t) \neq 0$. Teneral flies, $S_V$, are considered susceptible to *T. b. gambiense* infection with probability $p_V$ on their first blood meal, after this time the "teneral phenomenon" results in previously fed flies, $G_V$, having reduced susceptibility to infection by a factor $\varepsilon$ compared to unfed flies.

The proportion of tsetse bites taken on low-risk and high-risk humans are $f_1$ and $f_4$, depending on the relative abundance and proximity of the two risk groups. If $s_i$ is the relative availability of host type $i$ and high-risk humans are assumed to be $r$-fold more likely to receive bites, then $s_1 = 1$ and $s_4 = r$. Therefore, $f_i$'s can be calculated using $f_i = \dfrac{s_i N_{Hi}}{\sum_j s_j N_{Hj}}$.

**Improvements to passive case detection.** For simulations in this study, we assumed that prior to 1998 there was limited passive case detection, which would not detect stage 1 cases ($\eta_H^{\text{pre}} = 0$) and have a slower time to detection for stage 2 ($\gamma_H^{\text{pre}} = b_{\gamma_H^{\text{pre}}} \times \gamma_H^{\text{post}}$ where $b_{\gamma_H^{\text{pre}}} = [0, 1]$). In 1998 we assume that the introduction of the card agglutination test for trypanosomes (CATT) enabled better diagnosis and stage 1 and 2 rates were increased to $\eta_H^{\text{post}}$ and $\gamma_H^{\text{post}}$ respectively.

As explored in a previous modelling study focusing on former Bandundu province [16], there is strong evidence of improvements to the passive surveillance system during 2000–2012 from examining the changes to the proportion of passive cases identified in stage 1 compared to stage 2. The health-zone level data utilised here for the main model fitting do not contain staging information before 2015 (and generally there are relatively few cases in 2015 and 2016). The staged case information from 2015 and 2016 were aggregated to the former province level and used alongside the provincial-level staged case data for 2000–2012 [28] to set prior distributions for logistic functions which we use to describe improvements to both the passive stage 1 and stage 2 detection rates in each year:

$$\eta_H(Y) = \eta_H^{\text{post}} \left[ 1 + \frac{\eta_{H_{\text{amp}}}}{1 + \exp\left(-d_{\text{steep}}(Y - d_{\text{change}})\right)} \right] \qquad (2)$$

$$\gamma_H(Y) = \gamma_H^{\text{post}} \left[ 1 + \frac{\gamma_{H_{\text{amp}}}}{1 + \exp\left(-d_{\text{steep}}(Y - d_{\text{change}})\right)} \right] \qquad (3)$$

These functional forms were used in former Bandundu and Bas Congo provinces where there was strong evidence of improvement in the data. In Bas Congo the "change year", $d_{change}$, was fixed to 2015.5 as this corresponds to the year in which there was a substantial increase in the number of fixed health facilities with gHAT rapid diagnostic tests (RDTs) [29].

It is noted that only the stage 1 detection rate was changed in Model W in Castaño *et al.* [16] as this provided a good fit to the data at the province level. Here we alter both stage 1 and stage 2, firstly because it is logical that passive detection improvements would lead to faster rates for both stages of disease, and secondly as this is better able to capture passive detection patterns observed in health zone level data. In the present study other provinces were assumed to have constant passive detection rates since 1998.

**Active screening algorithms.** Active screening since 2000 has generally comprised mobile teams screening as many people as possible in villages using a multi-diagnostic algorithm. The first diagnostic used is the CATT test on finger prick blood, and this may be followed by CATT dilutions and finally microscopy to visually confirm presence of the parasite. After parasite confirmation, the individual is a confirmed case and required further testing using lumbar puncture to diagnose disease stage to be able to provide the correct stage-specific treatment. In the model we assume this algorithm has a sensitivity of 0.91, and very high but imperfect specificity with a prior around 0.9995 and fitted to health zone level data.

Whilst the national programme, PNLTHA, have consistently used the algorithm with parasitilogical confirmation, in former Oriental province in the North of the country, some of the screening activities (pre-2013) were performed by MSF. The MSF algorithm was more contracted and individuals who were found positive on a CATT 1:32 were reported as cases and given treatment. Therefore, a higher fixed sensitivity (MSF sensitivity = 0.95 in contrast to PNLTHA sensitivity = 0.91) and a lower fitted specificity (= $b_{specificity} \times$ specificity with targeted mean = 0.991, where $b_{specificity} = [0, 1]$) are used for years up to 2012 in these regions.

Since 2015, video confirmation of parasitological diagnosis was introduced to Mosango and Yasa Bonga health zones in former Bandundu province. This additional validation diagnostic is designed to ensure quality control of case confirmation, which is especially important as elimination is approached and very few cases remain. The model uses the assumption that there have therefore been no false positives in active screening since 2015 in Mosango and Yasa Bonga.

**Vector control.** Between 2000–2016 there were very limited vector control activities in DRC, with tsetse control implemented in a single health zone, Yasa Bonga in former Bandundu province, via the deployment of tiny targets since mid-2015. This method of control, using insecticidal impregnated blue targets, has successfully reduced fly populations in other countries (90% in Uganda [30], 80% in Guinea, and 99% in Chad). In the present study we incorporate vector control with a 90% reduction in tsetse for Yasa Bonga [27] (see SI "S1 Text" for further details).

## Likelihood

Eight parameters; $R_0$, $r$, $\eta_H^{post}$, $\gamma_H^{post}$, $b_{\gamma_H^{pre}}$, $k_1$, $u$, and *Spec* were fitted in all health zones. Additional parameters were included as required (combinations of $d_{change}$, $\eta_{H_{amp}}$, $\gamma_{H_{amp}}$, $d_{steep}$ and $b_{specificity}$ as appropriate, see above).

The Metropolis-Hastings MCMC used a log-likelihood function:

$$
\begin{aligned}
LL(\theta|x) \;=\;& \log\left(P(x|\theta)\right) \\
\propto\;& \sum_{t=2000}^{2016}\left(\log\left[\mathrm{BetaBin}\left(A_{D1}(t)+A_{D2}(t);z(t),\frac{A_{M1}(t)+A_{M2}(t)}{z(t)},\mathrm{disp_{act}}\right)\right]\right. \\
& + \log\left[\mathrm{Bin}\left(A_{D1}(t);A_{D1}(t)+A_{D2}(t),\frac{A_{M1}(t)}{A_{M1}(t)+A_{M2}(t)}\right)\right] \\
& + \log\left[\mathrm{BetaBin}\left(P_{D1}(t)+P_{D2}(t);N_{H},\frac{P_{M1}(t)+P_{M2}(t)}{N_{H}},\mathrm{disp_{pass}}\right)\right] \\
& \left.+ \log\left[\mathrm{Bin}\left(P_{D1}(t);P_{D1}(t)+P_{D2}(t),\frac{P_{M1}(t)}{P_{M1}(t)+P_{M2}(t)}\right)\right]\right)
\end{aligned}
$$

The model takes parameterisation $\theta$, $x$ is the data, $P_{Dj}(t)$ and $A_{Dj}(t)$ are the number of passive/active cases (of stage $j$) in year $t$ of the data, $P_{Mj}(t)$ and $A_{Mj}(t)$ are the number of passive/active cases (of stage $j$) in year $t$ of the model, and $z(t)$ is the number of people screened in year $t$. BetaBin$(m;n,p,\rho)$ gives the probability of obtaining $m$ successes out of $n$ trials with probability $p$ and overdispersion parameter $\rho$. The overdispersion accounts for larger variance than under the binomial. The pdf of this distribution is given by:

$$
\mathrm{BetaBin}(m;n,p,\rho)=\frac{\Gamma(n+1)\Gamma(m+a)\Gamma(n-m+b)\Gamma(a+b)}{\Gamma(n-m+1)\Gamma(n+a+b)\Gamma(a)\Gamma(b)}
$$

where $a=p(1/\rho-1)$ and $b=a(1-p)/p$.

Consequently larger $\rho$ yield more overdispersion. To avoid overfitting, the overdispersion parameters were left fixed at a value appropriate for a health zone level fit across MCMC runs (Table 1). The value of $\rho$ was chosen based on the median of the log posterior probability distribution achieved from MCMC runs with $\rho$ fixed at a range of values for two example health zones.

For four health zones there was no screening reported in one or two years in which there was more than 20 active cases recorded. In this scenario the number of negative test results in year $t$, $A_D^-(t)$, was sampled from a negative binomial distribution and $\hat{z}(i)=A_{D1}(t)+A_{D2}(t)+A_D^-(t)$ was used in the calculation of the log-likelihood in place of $z_t$. More detail is provided in the Supplementary Information (S1 Text).

Health zone results are aggregated by provinces and are compared to the province level data as a check that the health zone fits make sense.

## Priors

Prior distributions for the fitted parameters are given in Table 2 for the parameters fitted in all health zones and those fitted in former Bandundu province. Information on other former province–specific priors is given in the SI (S1 Text).

Informative priors were used for most parameters. There was little information on the stage of the disease in the data, essentially none before 2015. Former–province level staged case numbers were available from 2000–2012 [28] and we augmented these with our data aggregated to the former–province level for the years 2013 to 2016. The model was fitted to these staged former–province level data and informative priors for the parameters relating to treatment rates from stages 1 and 2 ($\eta_H^{\mathrm{post}}$ and $\gamma_H^{\mathrm{post}}$), and improvement in passive detection ($\eta_{H_{\mathrm{amp}}}$, $\gamma_{H_{\mathrm{amp}}}$ and $d_{\mathrm{steep}}$) were based on the respective posterior distributions from these analyses.

## Markov chain Monte Carlo algorithm

For each health zone the model was fitted using the adaptive Metropolis-Hastings random walk algorithm [31]. Two independent chains were run in parallel from different starting values. The chains were run in 3 phases: a transient phase, an adaptive phase and a sampling phase and only samples from the final phase were used in the analysis.

The aim of the transient phase was to move the chains towards the posterior mass. In this phase, lasting $B = 500$ iterations, single site parameter updates were used with proposal standard deviation $\sigma_n^i$ for parameter $i$ in proposal $n$. If proposal $n$ was accepted then $\sigma_{n+1}^i = 2\sigma_n^i$ and if it was rejected then $\sigma_{n+1}^i = 2^{a/(a-1)}\sigma_n^i$. Here, we target an acceptance rate of $a = 0.44$, which was found to be optimal in [32].

The aim of the adaptive phase was to begin learning the covariance matrix of the posterior in order to find an efficient proposal. In this phase proposals were drawn from $\boldsymbol{Y}_{n+1} \sim N_d\left(\boldsymbol{X}_n, \frac{2.38^2\lambda_n^2}{d}\boldsymbol{\Sigma}_n\right)$, where $\boldsymbol{X}_n$ is the location of the chain after $n$ iterations. The covariance matrix was initially $\Sigma_B$ (the initial variances used can be found in the parameter file which accompanies the code, https://doi.org/10.17605/osf.io/ck3tr) and subsequently

$$\boldsymbol{\Sigma}_{n+1} = \frac{n - B}{n - B + 10}\text{Cov}(\boldsymbol{X}_{B+1}, \ldots, \boldsymbol{X}_n) + \frac{10}{n - B + 10}\boldsymbol{\Sigma}_B. \tag{6}$$

The scaling factor was $\lambda_n = 1$ on even iterations and updated adaptively on odd iterations, namely $\lambda_{n+2} = x_n\lambda_n$ if iteration $n$ was accepted and $\lambda_{n+2} = x_n^{a/(a-1)}\lambda_n$ if rejected. We used $x_n = 1 + \frac{50}{50+n-B}$ and targeted an acceptance rate of $a = 23.4\%$, which was found to be optimal in [32].

The duration of the last two phases was determined adaptively, by examining the Gelman-Rubin convergence diagnostic [33] and the Effective Sample Size (ESS) diagnostic [34]. The adaptive phase was increased iteratively by 100 up to a maximum of $10^5$ iterations until the following convergence criteria were satisfied for the most recent 2000 observations. First $R_{\text{within}}^{(i,j)} < 1.1$ for every parameter $i$ and chain $j$; and second $R_{\text{between}}^{(i)} < 1.5$ for all $i$.

Finally, in the sampling phase, proposals were drawn as for the adaptive phase. The duration of this phase was determined based on ESS. To achieve a sample of 2,000 states with an ESS of at least 1,000, each chain was run for 1,000 iterations, thinned by a factor $\kappa = 1$ and the ESS calculated to be twice the minimum ESS over the parameters and the two chains. If the ESS criterion was not satisfied then a further 1,000 iterations were sampled from each chain and the thinning factor $\kappa$ was increased by 1, up to a maximum of 200. If the ESS criterion was still not satisfied then the health zone was flagged for investigation. If an analysis finished with ESS $\leq 500$ or $\max R_{\text{between}}^{(i)} \geq 1.2$ then that analysis was flagged for investigation, consisting of manual appraisal of the joint posterior distributions and progress to convergence. None of the analyses had $\max R_{\text{between}}^{(i)} \geq 1.2$, and there was no evidence that those with ESS $\leq 500$ were not converging correctly. Additional sampling was carried out for all analyses where ESS $< 1,000$, allowing $\kappa$ to exceed 200.

## Results

Fig 4 shows the final maximum values of the Gelman-Rubin convergence diagnostic for each health zone level analysis within former province. A good level of convergence was achieved, with only 1 of 168 analyses having $\max R_{\text{between}}^{(i)} > 1.02$ while 149 of the 168 analyses had $\max R_{\text{between}}^{(i)} \leq 1.01$.

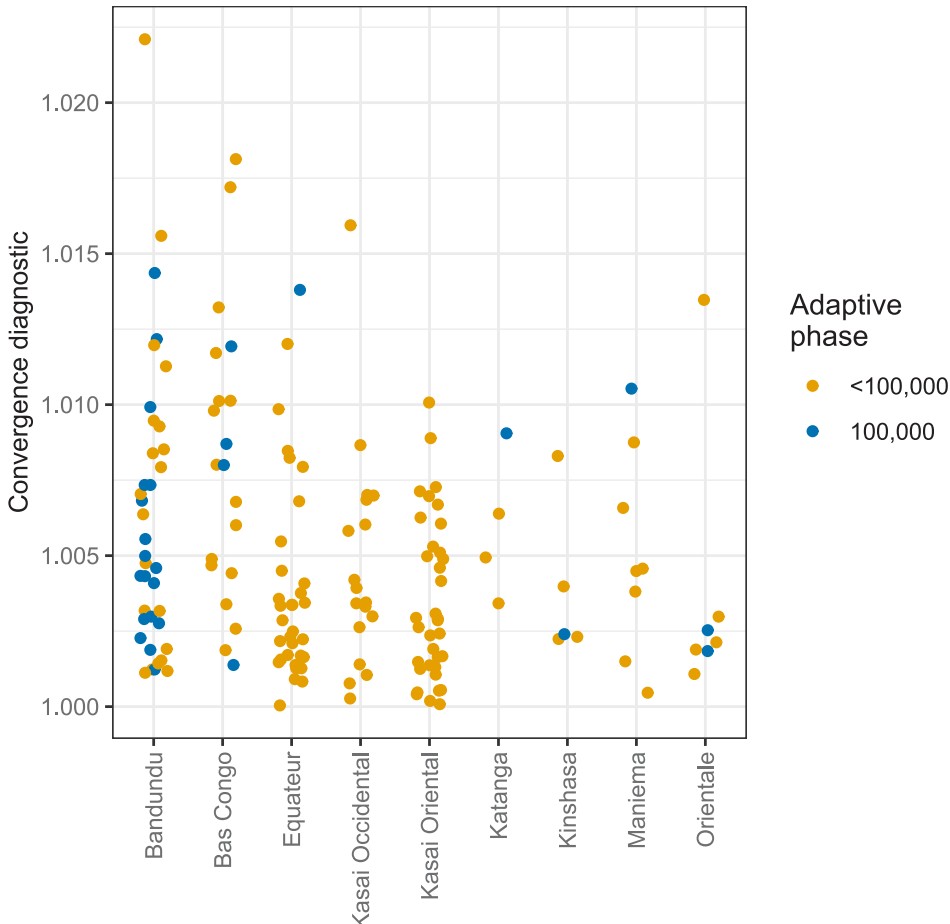

**Fig 4. Maximum between-chain Gelman-Rubin convergence diagnostics for the final posterior samples, grouped by former province and whether or not the maximum length of adaptive phase (100,000 iterations) was reached.**

The maximum number of iterations for the adaptive phase was required in 28 analyses. Of these, 18 were in the former province of Bandundu in which all health zone analyses included fitting of additional parameters to account for changes in the effectiveness of passive surveillance. It may be possible to improve the adaptive MCMC to reduce the number of analyses carrying out the maximum adaptive phase length, however the levels of convergence already achieved imply that the effort required may not result in markedly improved analysis results.

The observed effective sample size (ESS) exceeded 1,000 in 140 of the individual health zones analyses, and was less than 500 for 13 analyses. As with the adaptive phase, 14 of 28 analyses with an ESS less than 1,000 were for health zones in Bandundu. This again reflects the additional complexity of fitting the model in this former province.

The fitting process matches model outcomes to reported timeseries of actively- and passively-detected cases. Fig 5 shows examples of these trends for two example health zones; Kwamouth (in the former province of Bandundu) and Tandala (in the former province of Equateur). Kwamouth has a much higher incidence of gHAT infection than Tandala, and consequently has higher numbers of people being actively screened each year—the average number of people actively screened annually in Kwamouth was 52% of the estimated population in 2015 (127,205), while this was 6.8% for Tandala (estimated 2015 population of 274,945). Fig 5 shows how well the model fits to the timeseries of reported cases both actively and passively

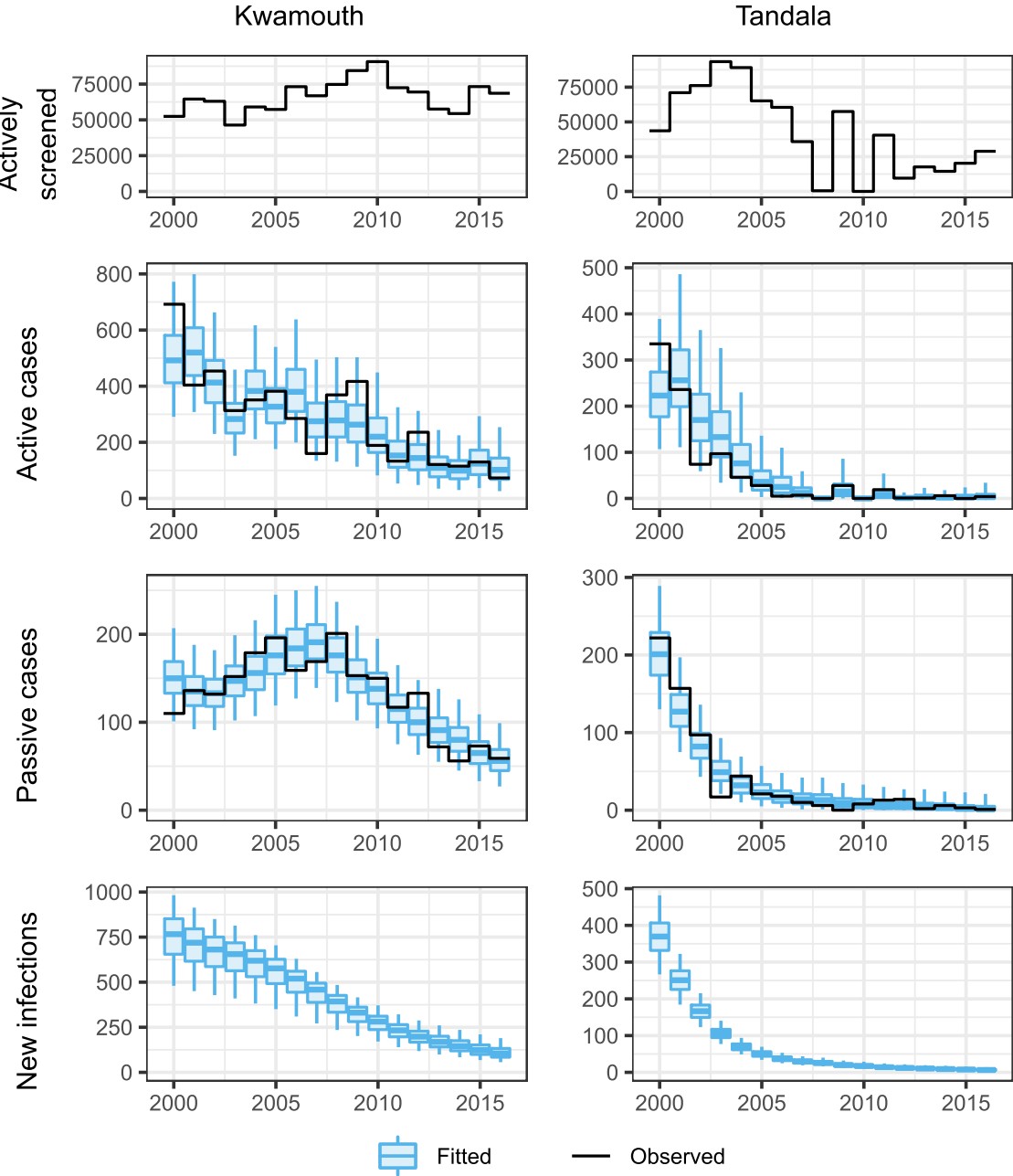

**Fig 5. Demonstration of fit to the observed trends in new case detection over time and predicted numbers of new infections for two example health zones; Kwamouth in former Bandundu province and Tandala in former Equateur province.**

detected. In addition, Fig 5 also shows the unobservable downward trend in new infections estimated by the model. The results of all 168 health zone level fits including inferences about annual numbers of new infections, are available online via a graphical user interface (https://hatmepp.warwick.ac.uk/fitting/v2/).

In Kwamouth health zone there is a noticeable "humped" trend in passive case detection, with higher case reporting in 2005–2008. The model is able to replace this trend by increasing the passive detection rates, $\eta_H(Y)$ and $\gamma_H(Y)$, and importantly we infer that underlying

**Table 3. Reduction in new gHAT infections by former province.** Medians and 95% credible intervals (CIs) of aggregated health zone-level outcomes.

| Former province | New infections (median [95% CI]) | | | Percentage reduction | |
|---|---|---|---|---|---|
| | **2000** | **2008** | **2016** | **2000–2008** | **2000–2016** |
| Bandundu | 7336 [6758, 7929] | 3186 [2917, 3433] | 801 [682, 954] | 57 [54, 59] | 89 [87, 91] |
| Bas Congo | 804 [721, 891] | 204 [181, 231] | 58 [45, 76] | 75 [72, 77] | 93 [90, 94] |
| Equateur | 4564 [4149, 4978] | 446 [405, 490] | 160 [136, 190] | 90 [89, 91] | 96 [96, 97] |
| Kasai Occidental | 700 [597, 835] | 313 [268, 364] | 163 [133, 198] | 55 [51, 59] | 77 [72, 81] |
| Kasai Oriental | 3534 [3232, 3880] | 1461 [1330, 1623] | 857 [746, 1021] | 59 [55, 61] | 76 [72, 79] |
| Katanga | 145 [109, 193] | 135 [102, 178] | 117 [85, 161] | 7 [-3,17] | 20 [-1,37] |
| Kinshasa | 238 [173, 322] | 135 [102, 184] | 85 [57, 125] | 43 [32, 53] | 64 [50, 75] |
| Maniema | 233 [191, 285] | 174 [147, 209] | 150 [121, 186] | 25 [17, 34] | 36 [22, 49] |
| Orientale | 2301 [2054, 2550] | 1528 [1324, 1742] | 739 [596, 897] | 33 [26, 41] | 68 [61, 74] |

transmission actually declines during this time period, despite increased case reporting (see Fig 5 and online results https://hatmepp.warwick.ac.uk/fitting/v2/). This humped shape of passive case reporting is observed in many health zones of former Bandundu province, such as Bokoro, Bolobo, Ipamu, and Yasa Bonga. Much of former Equateur province has a different typical pattern in its passive detection trend which looks similar to exponential decay, especially in the north (e.g. Bominenge, Boto, Budjala, Gemena, Karawa, Kungu, Libenge, Tandala) where there was very high case reporting in the early 2000s. In these locations we reproduce the trend using fixed passive case detection rates combined with successful active screening.

Table 3 contains former province-level estimates of the number of new infections in 2000, 2008 and 2016 and the percentage reduction in new infections between these years. The values from each of 1,000 posterior samples for each analysed health zone within a former province were summed to give 1,000 former province–level values, each health zone being independent in our analyses. Over this period, a 96% median reduction in the number of new infections has been achieved in Equateur. The largest percentage reductions in the number of new infections are seen in the former provinces with the highest reported cases in 2000. Notably the province with the highest reporting in 2000 (Equateur), was not estimated to have the highest transmission (Bandundu was higher), but does have a huge inferred reduction in transmission of 96%. Despite a calculated 89% transmission reduction, Bandundu province remained the province with most ongoing transmission in 2016.

The fits of the model to the reported case timeseries as exemplified in Fig 5 reflect the parameters of the model, many of which are estimated within the fitting process. Fig 6 illustrates the posterior distribution of $R_0$ within each health zone for which inference was performed. The map of DRC was partitioned into hexagons and partial hexagons, based on health zone boundaries, which were then filled with a colour based on a random value from the posterior distribution of $R_0$ for that health zone. By representing the posterior distributions in this way the aim is to illustrate how both the level and variability of the parameter differs between health zones. The geographical unit of interest here is the health zone and the locations of the hexagons within the healthzone are meaningless. Further examples of this representation of posterior distributions are available for the parameters fitted across all health zone analyses in the Supporting Information (S2 Text) and online https://hatmepp.warwick.ac.uk/fitting/v2/).

The $R_0$ posterior map shows that some of the highest $R_0$ estimates are in Kwamouth health zone, former Bandundu province, although more generally $R_0$ estimates remain very low—typically only slightly above one—for the whole country. $R_0$ is a bundled parameter, linked to the ability of an infection to persist for a particular geographic setting. Higher $R_0$ values

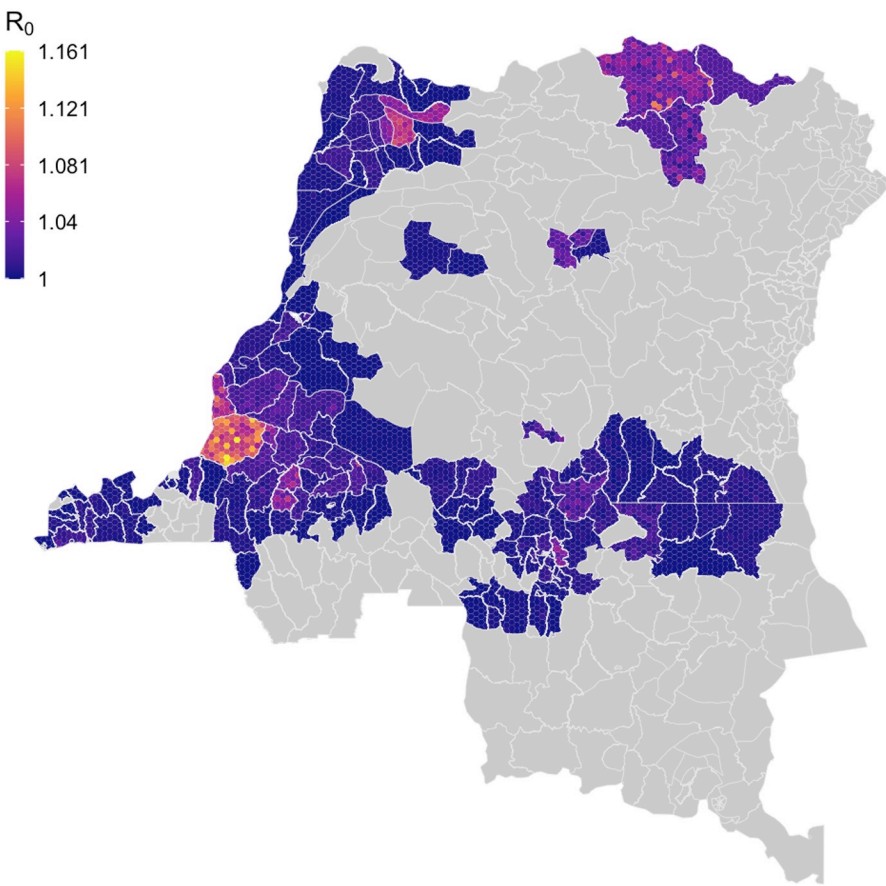

**Fig 6. Within health zone posterior distribution of $R_0$.** Fill colours for hexagons within a health zone are determined by randomly sampled values from the posterior distribution of $R_0$ from the analysis of that health zone. Shapefiles used to produce these maps are available under an ODC-ODbL licence at https://data.humdata.org/dataset/drc-health-data.

on the map roughly correspond to health zones which have reported high number of cases historically.

Maps showing posterior passive detection rates (see $\eta_H^{\text{post}}$ in SI; file S2, Fig S2.3) show that there is geographic variability in time to detection, with typically quicker diagnosis in 2000 in former Equateur province and slower times in former Orientale, Maniema and Katanga. The average time that a person spent infectious, if not identified by active screening, can be estimated in each health zone using the posterior parameters (see S3 Text); in Kwamouth health zone this duration changes from a median of 1224 days in 2000 to 813 days in 2016 and in Tandala this is fixed at 716 days between 2000 and 2016.

The specificity of the active screening diagnostic algorithm is computed to be very high, with former Equateur province having near perfect specificity. The parameter $u$ corresponds to the proportion of cases that are reported if not picked up in active screening. High $u$ is interpreted as high reporting and fewer deaths outside the health care system. Average reporting generally follows the prior of $u$ although it appears that former Equateur has better reporting (lower under-reporting/fewer deaths). The estimated proportion of reporting including active screening can be computed using model outputs for cases and deaths (see S3 Text). In Kwamouth this is estimated to have changed from 0.67 (95%: 0.54–0.84) in 2000 to 0.82 (95%: 0.62–0.93) in 2016, while in Tandala it fluctuated over time from 0.65 (95%: 0.53–0.75) in 2000

and finishing back at 0.65 (95%: 0.0–1.0) in 2016, with the median varying between 0.40 (in 2010) and 0.76 (in 2003). The low case numbers in later years are the cause of the higher uncertainty in the percentage reporting in 2016.

## Discussion

The fitting process presented in this study enabled exploration of the underlying epidemiology of gHAT at the health zone level, estimating parameters of the transmission model based on historical data. The fits highlight the success of past interventions, both in the obvious decline in the number of reported cases but also through quantification of improvements in surveillance, such as in the case of changes in passive surveillance over time in Bandundu and Bas Congo.

Even in areas where there has been a decline in active screening activity over time (especially in former Equateur province where the mean annual screening from 2012–2016 was ~150,000 compared to a maximum in 2003 of around 900,000), the modelling indicates that there has been a real reduction in transmission (~96% in Equateur), rather than simply a decline in reported cases due to scaling back the detection effort.

We note that the health zone-level longitudinal case data typically did not have staging information associated with them prior to 2015. The staging information for 2015 and 2016 were highly informative in quantifying passive detection improvements when combined with provincial-level staged data published separately [28]. We reiterate the recommendation of Castaño *et al.* [16], in collecting and digitising staged case data whenever possible to enable continued assessment of the passive surveillance system. The number and type of diagnostic tests used in passive surveillance would also be valuable in assessing the effectiveness and activity levels of the passive surveillance system.

There are other areas where additional data recording, either routine or as part of systematic surveys could be beneficial. For example, the age and gender profile of gHAT cases and populations screened would provide valuable information regarding high- and low-risk groups of individuals, and potentially also their participation in screening.

Vector control activities are expanding in DRC. In the data used here, vector control had only begun in 2015 in Yasa Bonga health zone and the model with respect to vector control was parameterised using a fixed value for the reduction in tsetse population taken from the entomological follow-up studies in this area. Incorporation of repeated entomological survey data and human tsetse exposure data as in, for example, Courtin *et al.* [35] may help the model be parameterised appropriately for each vector control region, incorporating uncertainty about the vector control-related parameters.

In this study we have been able to estimate the relative vector-to-host ratio for different regions through our adaptive MCMC. Maps of $m_{eff}$ are shown in SI "S2 Text", however these alone do not tell the whole story and are not necessarily reflective of where one might find the highest density of flies in DRC. In particular the high-/low-risk structure of the model (associated with parameters $k_1$, the proportion of the population who are low risk, and $r$ the relative exposure of high-risk individuals) is intimately entwined with the relative vector-to-host ratio. It would, therefore, be possible in this framework to have a comparatively low $m_{eff}$ in a health zone, but high $r$ due to dense pockets of tsetse habitat which only high-risk people are exposed to; overall this could lead to a high burden health zone due to population-level heterogeneity in risk.

Despite the complex nature of inferring the role of tsetse, we do examine a single case where deliberate vector control has been deployed during the time period of our dataset. In Yasa Bonga, tiny targets were used from 2015 and although it is not strongly apparent by

simply looking at case detections in 2016, we estimate that transmission in the region has sharply fallen from around 22.6 (95% CI: 11.2–41.2) new infections per year in 2015 to 1.3 (95% CI: 0.6–2.5) in 2016. Due to the slow progression of the disease, it would be expected to take several years of case data until there is a marked decline in both active and passive case detections. In future studies, more years of human case and tsetse monitoring data from Yasa Bonga and other health zones, which have subsequently begun tsetse control, will help to provide deeper insight into the role of these vectors in transmission dynamics.

The model used here did not consider the presence of possible animal reservoirs or transmission via asymptomatic humans. Historically, gHAT has been generally regarded as an anthroponosis [36], lacking an animal reservoir which would hinder or prevent elimination of the disease. More recently there have been concerns that this may not be the case; *T.b. gambiense* has been identified in various animals, and have been shown to be transmissible to tsetse experimentally [37]. Mathematical modelling has attempted to ascertain the likelihood of animal transmission through various model comparison exercises [6, 7], and also quantify relative transmission if it occurs. To date there is inconclusive evidence, although declines in transmission due to medical intervention over the last decades appear to rule out substantial animal transmission (see [38] for more discussion on this topic). A more detailed analysis of the presence and/or role of animal reservoirs is beyond the scope of the present study.

The gHAT model presented is an explicitly Ross-Macdonald-style, host-vector model capable of simulating the impact of vector population size changes on transmission to humans. This was introduced to enable the simulation of intentional vector control which was deployed in Yasa Bonga health zone from 2015 and so that future modelling using this framework could explore the possible impact of vector control in other locations. We did not, however, consider other sources of fluctuation of tsetse populations, although it is noted that anthropogenic change in particular could result in loss of tsetse habitat and inadvertently reduce tsetse populations [39]. There are limited data available on temporal changes in tsetse density across DRC to inform such an analysis, however it could be a potential avenue for future research in areas of known deforestation or urbanisation.

One published review [40] has previously suggested that the primary transmission mechanism of gHAT is not tsetse. In the present modelling study, for the most part, it would be unlikely to make any difference to our results to assume human-to-human transmission as we assume stable vector populations which could be readily substituted by a quasi-equilibrium assumption or even modelled as a contagion without substantial quantitative differences. However, where vector control is implemented, our assumption that gHAT is a vector-borne infection would produce very different results compared to assuming no or limited tsetse transmission. This would be very pronounced if the model is subsequently used to make projections of future vector control impact. We are reassured by various other recent studies that our tsetse-transmission assumption is valid as (i) the host-vector gHAT model predictions for regions with vector control appear to match the case reductions well [7] and (ii) the natural experiment which occurred in Guinea during the 2014–15 West African Ebola outbreak found that even following interruption of medical screening activities, regions that had vector control remaining in situ had much lower case burden following resumption of screening than those without [41, 42].

The model framework utilised in the present study is deterministic, always yielding the same outputs for the same set of input parameters—deterministic models can be considered to represent expected average dynamics. Whilst we do address some observational uncertainty through the overdispersion in the likelihood function and drawing model case reporting outputs, the use of stochastic models which capture chance events in transmission and reporting will become increasingly important as infections approach zero. Other modelling studies

utilising stochastic model formulations have found that, even at low reported case numbers and for relatively small populations (>2000 people), gHAT persists with high probability over long periods [5]. At the health zone level (∼100,000 people) model dynamics follow closely to deterministic ones until reaching extremely low case numbers [16]. We therefore consider that the deterministic model fits presented here would be similar to those obtained using a stochastic model variant in health zones with appreciable case reporting in the 2000s and regular good-coverage active screening. In health zones with limited case detection despite good screening coverage, a stochastic model would be more appropriate to obtain robust model fits. Future analysis should consider how the adaptive MCMC framework presented here could be built upon to fit stochastic model variants, especially for fits at smaller geographic scales (e.g. health areas ∼10,000 people) are required.

## Conclusions

The gHAT model and automated adaptive MCMC approach presented here has facilitated the fitting of longitudinal case data across the whole DRC for the first time. The flexible framework will support future studies with straightforward fitting of other model variants (such as those with animal reservoirs or asymptomatic human infection) or to updated data sets in DRC or elsewhere.

The results of the fitting suggest that dwindling case reporting in many parts of the country does correspond to a real reduction in underlying transmission, even in locations where active screening coverage has also declined; Equateur province is a prime example of this. The passive detection posterior parameters found for Bandundu province indicate that substantial improvements have been made to reduce time to detection in the region since the turn of the century—we are now able to quantify the increase in the proportion of infections that are eventually reported rather than die undiagnosed for each health zone.

These country-wide analyses have given an insight into why transmission may persist in some locations more than others, for example the results indicate that passive detection rates were higher in Equateur province in the early 2000s than in Bandundu province and teamed with active screening this resulted in a more marked decline in both cases and transmission. As well as interventions, epidemiological factors, including human behaviour, are likely driving infection differences: the relative risk of high-risk people compared to low-risk appears to be greater in known persistent regions of Bandundu—including the health zones of Kwamouth, Bolobo, Mushie and Bagata—compared to other provinces.

This finding raises the question of whether current medical-only interventions are sufficient to reduce transmission to the point of interruption before 2030 in all regions, or whether other approaches may be needed in locations with moderate but less substantial reductions. Indeed, previous modelling for specific health zones in DRC has suggested intensified interventions such as targeting high-risk groups in active screening or vector control may be needed to speed up progress and meet the 2030 goal [13, 15]. A larger challenge is to identify where this may be needed across the country, and where should be prioritised for bolstered interventions.

This study is a necessary first step towards providing modelling information which can assist in the formulation of policy appropriate to the varying needs across the country. We have followed the five principles set out and recommended for good modelling practise—known as Policy-Relevant Items for Reporting Models in Epidemiology of Neglected Tropical Diseases (PRIME-NTD) [43] (see S4 Text). Whilst beyond the scope of the present study, the fits obtained here may now be used to simulate projections of the disease into the future under various different interventions. Once forecasts have been made, the results can be used to

examine minimal strategies to achieve the 2030 elimination of transmission goal and economic modelling can be used to ascertain location-specific, cost-effective options, refining more general health economic analyses presented previously [44].

## Supporting information

**S1 Text. Materials and methods.** More detailed description of materials and methods.
(PDF)

**S2 Text. Posteriors of fitted parameters.** Additional results figures and tables. Representations of posterior distributions of fitted parameters for the two example health zones, across the country and within the former provinces of Bandundu and Bas Congo.
(PDF)

**S3 Text. Using posteriors to infer time infected and reporting.** Calculating the average time spent infected and the proportion of infections that are reported.
(PDF)

**S4 Text. PRIME-NTD criteria.** Addressing the PRIME-NTD criteria for good modelling practises.
(PDF)

## Acknowledgments

The authors thank PNLTHA for the original data collection, WHO for data access (within the framework of the WHO HAT Atlas [1]), and Cyrus Sinai and Nicole Hoff from UCLA Fielding School of Public Health for providing health zone-level shapefiles (current versions can be found at https://data.humdata.org/dataset/drc-health-data).

## Author Contributions

**Conceptualization:** Simon E. F. Spencer, Matt J. Keeling, Kat S. Rock.

**Data curation:** Ronald E. Crump, Simon E. F. Spencer, Erick Mwamba Miaka, Chansy Shampa.

**Formal analysis:** Ronald E. Crump, Kat S. Rock.

**Funding acquisition:** Simon E. F. Spencer, Matt J. Keeling, Kat S. Rock.

**Investigation:** Ronald E. Crump.

**Methodology:** Edward S. Knock, Simon E. F. Spencer, Kat S. Rock.

**Project administration:** Simon E. F. Spencer, Kat S. Rock.

**Software:** Ching-I Huang, Edward S. Knock, Simon E. F. Spencer, Paul E. Brown, Kat S. Rock.

**Supervision:** Simon E. F. Spencer, Matt J. Keeling, Kat S. Rock.

**Visualization:** Ronald E. Crump, Ching-I Huang, Paul E. Brown.

**Writing – original draft:** Ronald E. Crump, Ching-I Huang, Simon E. F. Spencer, Kat S. Rock.

**Writing – review & editing:** Ronald E. Crump, Ching-I Huang, Edward S. Knock, Simon E. F. Spencer, Paul E. Brown, Erick Mwamba Miaka, Chansy Shampa, Matt J. Keeling, Kat S. Rock.

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
