## [Decision Letter · Decision Letter 0]

10 Aug 2020

Dear Dr Crump,

Thank you very much for submitting your manuscript "Quantifying epidemiological drivers of gambiense human African Trypanosomiasis across the Democratic Republic of Congo" for consideration at PLOS Computational Biology. As with all papers reviewed by the journal, your manuscript was reviewed by members of the editorial board and by several independent reviewers. The reviewers appreciated the attention to an important topic. Based on the reviews, we are likely to accept this manuscript for publication, providing that you modify the manuscript according to the review recommendations.

Sincerely,

Alex Perkins

Associate Editor

PLOS Computational Biology

Virginia Pitzer

Deputy Editor

PLOS Computational Biology

[LINK]

Reviewer's Responses to Questions

**Comments to the Authors:**

Reviewer #1: Please see attachements containing review and marked up copy of the MS

Reviewer #2: Summary:

This article presents a mechanistic model of Gambiense human African trypansomiasis transmission that is calibrated to case data at the health zone level from the Democratic Republic of Congo. The model is used to infer site-specific unobserved transmission parameters aimed at identifying geographical heterogeneities that may be important to understand persistence, especially in the context of eliminating transmission. This research is an important development for supporting elimination efforts for this neglected tropical disease, which is targeted for elimination by 2030. The paper is well-written and the motivation for the study is clear.

Major points:

1. It would be helpful to provide more detail regarding the manual tuning of the overdispersion parameters (lines 219-221). Was there a certain criterion used to deem the fit appropriate?

2. Was the health zone level data that was used for model fitting independent from the province level data described on lines 226-227? The data described in the Data section doesn’t refer to province level data, so I am wondering if this is a separate dataset used for validation purposes, or if the health zone and province level data come from the same raw dataset and this step was more of an internal check for consistency.

3. Similarly, there is reference to model fitting to province-level data on lines 234-235. Is this a distinct dataset? If not, what is the rationale for fitting at both the province and health zone levels?

4. I am curious about the rationale for choosing a single random value from the posterior distribution of R0 for showing geographical variation (line 318-322, Figure 6). Would a standardized sample across all locations, such as taking the median of the posteriors, be more representative?

5. The discussion surrounding model limitations and future applications is really nice, but it would be great for more to be included about the outcomes of the modeling done in the paper. The takeaway message and major findings could be stronger in this section. There were some interesting analyses done in this work that were presented well in the Results section and could be better highlighted in the Discussion.

Minor points:

1. Overall, the data is explained clearly, and the model structure and calibration procedure are well described. As it is currently described in lines 126 and 127, it was not immediately clear to me that the population was divided into low- and high-risk human categories. I interpreted this as low- and high-risk scenarios until I looked at Table 2 more closely. It might be helpful to draw attention to the parameters k1 and k4 in the text.

2. Since non-reservoir animals are not explicitly modeled, the non-reservoir animal compartment in Fig 3 could be removed for clarity.

3. Was there intended to be a conclusion section? There is currently a header with no content.

4. The paper could benefit from proofreading for phrasing and grammar. There are a few run-on and awkwardly phrased sentences, but overall, the paper is nicely written and organized.

Reviewer #3: Review uploaded as attachment

**Have all data underlying the figures and results presented in the manuscript been provided?**

Reviewer #1: Yes

Reviewer #2: Yes

Reviewer #3: Yes

PLOS authors have the option to publish the peer review history of their article (what does this mean?). If published, this will include your full peer review and any attached files.

Reviewer #1: **Yes: **John Hargrove

Reviewer #2: No

Reviewer #3: **Yes: **Luc E. Coffeng
---

## [Decision Letter · Decision Letter 1]

12 Nov 2020

Dear Dr Crump,

We are pleased to inform you that your manuscript 'Quantifying epidemiological drivers of gambiense human African Trypanosomiasis across the Democratic Republic of Congo' has been provisionally accepted for publication in PLOS Computational Biology.

Best regards,

Alex Perkins

Associate Editor

PLOS Computational Biology

Virginia Pitzer

Deputy Editor

PLOS Computational Biology

Reviewer's Responses to Questions

**Comments to the Authors:**

Reviewer #2: The authors have made substantial improvements to the manuscript and have addressed my comments. The methods section is more detailed and the main outcomes have been clarified. The discussion section has been expanded and reads very nicely.

Reviewer #3: I thank and congratulate the authors for carefully addressing all my comments.

One minor point on line 140: ordinary different equations - "differential" or "difference"?

**Have all data underlying the figures and results presented in the manuscript been provided?**

Reviewer #2: Yes

Reviewer #3: Yes

PLOS authors have the option to publish the peer review history of their article (what does this mean?). If published, this will include your full peer review and any attached files.

Reviewer #2: No

Reviewer #3: **Yes: **Luc E. Coffeng

---

## [Editor Report · Acceptance letter]

25 Jan 2021

PCOMPBIOL-D-20-01092R1 

Quantifying epidemiological drivers of *gambiens* human African Trypanosomiasis across the Democratic Republic of Congo

Dear Dr Rock,

I am pleased to inform you that your manuscript has been formally accepted for publication in PLOS Computational Biology. Your manuscript is now with our production department and you will be notified of the publication date in due course.

With kind regards,

Alice Ellingham
